# Pre-hospital causes for delayed arrival in acute ischemic stroke before and during the COVID-19 pandemic: A study at two stroke centers in Egypt

**Hany M. Aref, Hossam Shokri**[ID]**\*, Tamer M. Roushdy, Fatma Fathalla**[ID]**, Nevine M. El Nahas**[ID]

Neurology Department, Faculty of Medicine, Ain Shams University, Cairo, Egypt

\* hossam.shokri@med.asu.edu.eg

## Abstract

### Background

In the current study we investigated the causes of pre-hospital delay as this can compromise the patient's chance to receive thrombolytic therapy and thus impact stroke outcome.

### Methods

We surveyed 254 patients regarding reasons for delayed and early arrival to hospital after acute ischemic stroke. The survey was performed over five months, spanning a period pre- and during COVID-19 (between December 7, 2019 and May 10, 2020).

### Results

A total of 71.2% of patients arrived beyond four hours of onset of ischemic stroke. The commonest cause for delay pre-Covid-19 was receiving treatment in a non-stroke hospital, while that during COVID-19 was fear of infection and lock down issues. Not realizing the urgency of the condition and stroke during sleep were common in both periods. Early arrival because of the patient's previous experience with stroke accounted for approximately 25% of cases in both periods. The effect of media was more evident during COVID-19, accounting for 47.7% of cases.

### Conclusion

Pre-hospital delay secondary to misperception of the urgency of stroke and management in a non-stroke hospital reflect the lack of awareness among the public and medical staff. This concept is emphasized by early arrival secondary to previous experience with stroke and the pronounced effect of media in the time of COVID-19.

**Data Availability Statement:** All relevant data are within the paper and its Supporting Information files.

**Funding:** The author(s) received no specific funding for this work.

**Competing interests:** The authors have declared that no competing interests exist.

## Introduction

The concept of "time is brain" has been introduced over the past two decades to highlight the importance of salvaging brain tissue as early as possible after acute ischemic stroke (AIS) [1]. Early management of AIS is a multifactorial process that includes in-hospital as well as pre-hospital factors.

Before 2015, only 13.2% of eligible patients arriving to our hospital received reperfusion therapies for stroke [2]. It was found that 66.7% of patients arrived beyond the time window for thrombolysis. This pre-hospital delay was attributable to a lack of knowledge about the existence of intervention for acute stroke in 71.7% of cases, while in-hospital delay was mostly because of unavailability of recombinant tissue plasminogen activator (rtPA) in 56.5% of cases. Accordingly, in 2016, an action plan was implemented in order to improve in-hospital factors. This action plan resulted in reimbursement for thrombolysis services by Ministry Of Health (MOH) in 2016. Consequently, the rate of reperfusion therapy administered increased to 94.6% in eligible patients, and door to needle time was reduced from 68 minutes to 40 minutes, together with improvement in patients' outcomes [2–5].

However, the improvement in in-hospital stroke service was not accompanied by an equal reduction in pre-hospital delay. Consequently, only 16% of the total number of stroke patients presenting to our hospitals received thrombolytic therapy [3]. Similar findings have been reported by other centers that attribute the prolongation of "onset to needle time" to pre-hospital delay [6, 7].

COVID-19 has impacted stroke care in different countries. In some areas, many stroke units were re-organized and neurologists re-allocated to treat COVID-19 patients, while others adopted a stroke triage protocol to maintain an effective stroke service. However, most centers reported delayed patient arrival beyond the window for acute treatment that was attributed to fear of infection [8, 9].

Consequently, in the current study we attempted to explore causes of pre-hospital delay that interfere with optimum stroke service. This can be the first step in planning strategies for modifying these causes and further increasing the rate of utilization of thrombolytic therapy; thus, improving patients' outcomes.

Previous studies have investigated causes of pre-hospital delay; however, to the best of our knowledge this is the first study that also explores reasons for delay during a pandemic period.

## Methods

This study received approval from the Ethics Committee at Ain Shams University, Faculty of Medicine. This is a cross-sectional survey of a prospective cohort of all acute stroke patients admitted between December 7, 2019 and May 10, 2020. All patients were diagnosed clinically and confirmed by diffusion-weighted magnetic resonance imaging (MRI) study. The study was conducted after the approval of the hospital and faculty of medicine IRB. Patients were recruited from two stroke centers of Ain Shams University Hospitals in Cairo. Both are accredited stroke centers [10] and serve a catchment area of approximately 6.5 million people.

Within two days of admission, a survey (Table 1) was administered to the patient, or relative who brought the patient to hospital if the patient's condition would not allow. Inclusion criteria were all types of stroke, subjects who agreed to give informed consent, and those who were certain of the time of stroke onset. Patients were excluded if they were reluctant to complete the questionnaire or were unsure of the data requested.

The survey was conducted through a unified questionnaire and was performed under supervision of two authors (Table 1). Patients were assigned to two groups: delayed or early hospital arrival. The former group was asked about causes of delayed arrival and the latter about causes

**Table 1. Prehospital delay questionnaire.**

| | | | |
|---|---|---|---|
| **Patients name** | | | |
| **Patient ID** | | | |
| **age** | | | |
| **Gender** | male | female | |
| **Date of stroke onset** | | Time of stroke onset | |
| **Date of hospital arrival** | | Time of hospital arrival | |
| **Onset to door in min.** | | | |
| **Residency** | rural | Urban | |
| **Average distance from hospital in km.** | | | |
| **Type of stroke** | Ischemic | hemorrhagic | |
| **NIHSS on admission** | | | |
| **Type of management** | Reperfusion (rtPA) | Thrombectomy | conservative |
| **Education of care giver** | illiterate | Read and write | High school |
| | university | postgraduate | |
| **Education of patient** | illiterate | Read and write | High school |
| | university | postgraduate | |
| **Risk factors** | | | |
| | Smoking | yes | no |
| | DM | yes | no |
| | Hypertension | yes | no |
| | Dyslipidemia | yes | no |
| | Ischemic heart | yes | no |
| | Previous stroke | Yes | no |
| **Degree of orientation regarding stroke symptoms** | oriented | Not oriented that these symptoms belong to stroke | |
| **Causes of delay in patients come to hospital after 4 and half hours** | | | |
| | Waiting for symptoms to go away | | |
| | Not realizing the urgency of seeking medical help | | |
| | Stroke while sleeping | | |
| | Not able to call for help | | |
| | Taking medicine and waiting for it to take effect | | |
| | Calling doctor to come to the home | | |
| | Seek medical help at another hospital | | |
| **Causes of early arrival to hospital in patients come within window** | | | |
| | Oriented with stroke symptoms from media (type: TV. Radio. Social media, newspaper) | | |
| | Oriented with stroke symptoms previous experience self or relative | | |
| | Oriented with therapeutic window from media (type: TV. Radio. Social media, newspaper)) | | |
| | Oriented with therapeutic window previous experience (self, acquaintance) | | |
| | Referral from physician clinic | | |
| | Referral from hospital | | |
| | Knows the presence of hospital with stroke service nearby | | |
| **Method of transportation** | | | |
| | Private car | | |
| | Ambulance | | |
| **Referred from other hospital to us** | Yes | | no |
| **Received any ttt prior to arrival** | Yes | | no |

of early arrival. The mode of transportation to hospital was reported. Additionally, time of hospital arrival, risk factors, and severity of stroke measured by the National Institute of Health Stroke Scale (NIHSS) were reported from the SITS registry adopted by our stroke centers.

*The time of stroke onset* was defined as the time when the patient or relative first noticed neurological deficits suggestive of stroke. *The time of arrival to hospital* was defined as the time the physician at our stroke center first encountered the patient. *The time delay* was the difference between onset and arrival [11], and *delayed hospital arrival* was defined as greater than four hours from stroke onset.

The initial design of this study was to investigate the causes of delayed versus early hospital arrival in general. However, the COVID-19 pandemic ensued during the study, so the aim was extended to include a comparison between causes pre- and during the pandemic. The first case of COVID-19 was announced by the Ministry of Health in Egypt on February 15, 2020; therefore, two time intervals were studied: December 7 to February 14 (pre- COVID-19) and February 15 to May 10 (post- COVID-19).

Statistical analysis was done using SPSS© version 16 (SPSS Inc., Chicago, USA). The Shapiro-Wilks test was performed to test the normality of continuous data distribution. Continuous data (age, onset to door time, admission NIHSS and average distance from hospital) were presented as median and range for skewed data, whereas categorical data (gender, time of arrival [early versus late arrival], education of care giver and patient, hypertension, diabetes, dyslipidemia, smoking, ischemic heart disease, previous stroke, method of transportation, type of stroke and type of management) were presented as frequencies. Regarding bivariate analysis, Mann–Whitney U test and Kruskal–Wallis test were used to compare continuous variables that were not normally distributed with nominal independent variables (time in relation to COVID-19 versus age, onset to door, admission NIHSS and average distance from hospital), method of transportation versus onset to door, and education of care giver and patient versus onset to door time). Chi square test was used for comparison of nominal data (time in relation to COVID-19 versus gender). Fisher's exact test was used if >20% of the cells in any crosstabulation had an expected count of $\leq 5$ (time in relation to COVID-19 versus type of stroke, type of management, causes of pre-hospital delay and causes of early hospital arrival within window). Pearson's correlation coefficient was used to measure the linear correlation between two continuous variables (admission NIHSS versus onset to door time). $P < 0.05$ was considered statistically significant.

## Results

### Descriptive analysis of the studied sample: (Table 2)

A total of 304 patients were admitted through the emergency department during the entire study period (December 7 to May 10). Fifty patients were excluded either due to unidentified time of stroke onset or refraining from participation in the study (33 patients in the pre-COVID and 17 during COVID period). This study included 254 patients. In the pre-COVID period (69 days), 118 cases (46.5%) were recruited with an average of 13 patients per week, while in the COVID period 136 cases (53.5%) were recruited (85 days), with an average of 12.5 patients per week. The age range for the entire group of 254 patients was 55–68 years with a median of 61 years, and males representing 60.6%.

Ischemic stroke was the main type of stroke in 87% of patients, while 11.8% of patients presented with hemorrhagic stroke, and 1.2% presented with transient ischemic attack (TIA). The most frequent risk factor was hypertension in 63% of patients, while 49.2% had diabetes, 32.7% were smokers, 29.5% had ischemic heart disease, 22.4% had dyslipidemia, and 20.1% reported a previous stroke.

The number of patients with delayed arrival was 181, while those with early arrival was 73. The overall median onset to door time was 480 minutes with a range of 180–1440 minutes. Median NIHSS was 8 (5–10). Distance from the hospital had a median of 20 kilometers.

**Table 2. Clinical characteristics of the study subjects and comparison before and during COVID-19 time.**

| Variables | Number = 254 | | |
|---|---|---|---|
| | Frequency (percentage) | | |
| Age (years)* | 61 (55–68) | | |
| Males** | 154 (60.6) | | |
| Delayed / early arrival** | 181 (71.3) /73 (28.7) | | |
| Education of care giver** | | | |
| Illiterate | 43 (16.9) | | |
| Read and write | 55 (21.7) | | |
| High school | 68 (26.8) | | |
| University | 32 (12.6) | | |
| Postgraduate | 56 (22) | | |
| Education of patient** | | | |
| Illiterate | 60 (23.6) | | |
| Read and write | 63 (24.8) | | |
| High school | 62 (24.4) | | |
| University | 25 (9.8) | | |
| Postgraduate | 44 (17.3) | | |
| Hypertension** | 160 (63) | | |
| Diabetes** | 125 (49.2) | | |
| Dyslipidemia** | 57 (22.4) | | |
| Smoking** | 83 (32.7) | | |
| Ischemic heart disease** | 75 (29.5) | | |
| Previous stroke** | 51 (20.1) | | |
| Private transportation vs. Ambulance** | 234 (92.1%), 20 (7.9%) | | |
| | **Before COVID-19** | **During COVID-19** | **P VALUE** |
| | **N = 118 (46.5%)** | N = 136 (53.5%) | |
| Age (years)* | 60 (55–68) | 62 (54–68) | **0.56** |
| Male** | 61 (51.7) | 93 (68.4) | **0.007** |
| Onset to door (minutes)* | 540 (180–2160) | 480 (180–1215) | **0.29** |
| Type of stroke** | | | **0.002** |
| ischemic | 96 (81.4) | 125 (91.9) | |
| hemorrhagic | 22 (18.6) | 8 (5.9) | |
| TIA | 0 (0) | 3 (2.2) | |
| Type of management** | | | **0.38** |
| conservative | 98 (83.1) | 101 (74.3) | |
| rtPA | 17 (14.4) | 31 (22.8) | |
| rtPA and thrombectomy | 2 (1.7) | 3 (2.2) | |
| thrombectomy | 1 (0.8) | 1 (0.7) | |
| Admission NIHSS* | 6 (4–9) | 8 (6–11) | **<0.001** |
| Average distance from hospital (km)* | 20 (10–31) | 20 (10–30) | **0.35** |

*median (interquartile range).

**no. (percentage).

As for the type of management, 78.3% received conservative medical treatment in the form of antiplatelets, while 21.7% received revascularization therapy.

The level of education varied among caregivers and patients. Among caregivers, 26.8% received high school education, 22% had post graduate degree, 21.7% could read and write, 16.9% were illiterate, and 12.6% were university graduates. Among patients, 24.8% could read

and write, 24.4% had high school education, 23.6% were illiterate, 17.3% had postgraduate degree, and 9.8% were university graduates.

Transportation by private car was the predominant mode of transfer in up to 92.1% of cases.

## Comparison of patients' characteristics before COVID-19 and during COVID-19

The demographics of the study population were analyzed comparatively before and during COVID-19. There was no significant difference in age of patients presenting in either period. However, there was a remarkable preponderance of ischemic stroke and TIAs during COVID with less hemorrhagic events. The median onset to door time was less during COVID time than pre-COVID (480 minutes versus 540 minutes, respectively). Total cases receiving reperfusion therapy were greater during COVID (25.7%) versus pre-COVID (17.1%). The degree of severity was significantly more in the COVID time than before COVID, with a median of 8 and 6, respectively. P < 0.001 (Table 2).

## Causes of pre-hospital delay before and during the COVID-19 pandemic

Comparison of causes of pre-hospital delay before and during the COVID-19 pandemic showed that fear of COVID and delay because of lock down issues were two novel causes, representing 12% and 8.7%, respectively. When combined, both causes accounted for 20.7% i.e., approximately 20% of causes in the COVID period; thus, other causes seemed to be relatively reduced. Meanwhile, the most common cause of delay in the pre-COVID period was receiving medical help in another hospital before coming to our center (27%), while during the COVID period this cause came in fourth place (15.2%). Waiting for symptoms to go away, not realizing the urgency of the condition, and stroke during sleep were important causes in both periods (Table 3).

## Causes of early hospital arrival within window before and during COVID-19 pandemic

On the other hand, during COVID time, patients/relatives gave more positive responses to questions about knowledge of stroke symptoms. This increased orientation was ascribed to

**Table 3. Causes of pre-hospital delay before and during COVID-19 pandemic.**

|  | Before COVID-19 (N = 89) | During COVID-19 (N = 92) | P value |
|---|---|---|---|
| Stroke while sleeping (%) | 16 (18) | 18 (19.6) | 0.8 |
| Not realizing the urgency of seeking medical help (%) | 20 (22.5) | 16 (17.4) | 0.3 |
| Waiting for symptoms to go away (%) | 14 (15.7) | 15 (16.3) | 0.9 |
| Receiving medical help in another hospital* (%) | 24 (27) | 14 (15.2) | 0.05 |
| COVID fear (%) | NA | 11 (12) | NA |
| Transportation / lock down* (%) | NA | 8 (8.7) | NA |
| Taking medicine and waiting for it to take effect (%) | 8 (9) | 5 (5.4) | 0.3 |
| Not able to call for help (%) | 4 (4.5) | 3 (3.3) | 0.7 |
| Calling doctor to come to the home (%) | 3 (3.4) | 2 (2.2) | 0.6 |

*refers to non-stroke medical service.

NA = not appropriate.

**Table 4. Causes of early hospital arrival within the window before and during COVID-19 pandemic.**

| | Before COVID-19 (N = 29) | During COVID-19 (N = 44) | P value |
|---|---|---|---|
| | Frequency (%) | Frequency (%) | |
| Oriented with stroke symptoms from media (type: TV, radio, social media, newspaper) | 3 (10.3) | 21 (47.7) | <0.001 |
| Oriented with stroke symptoms, previous experience self or relative | 7 (24.1) | 11 (25) | 0.8 |
| Oriented with therapeutic window from media (type: TV, radio, social media, newspaper) | 1 (3.4) | 1 (2.3) | 0.5 |
| Oriented with therapeutic window previous experience (self, acquaintance) | 3 (10.3) | 2 (4.5) | 0.07 |
| Referral from physician clinic* | 4 (13.8) | 4 (9.1) | 0.2 |
| Referral from hospital* | 8 (27.6) | 2 (4.5) | <0.001 |
| Knows the presence of hospital with stroke service nearby | 3 (10.3) | 3 (6.8) | 0.3 |

*Referral here indicates immediate referral without attempting any management.

information from television, social media, and newspapers. Therefore, causes of early hospital arrival revealed significantly more orientation about stroke symptoms from media during COVID time compared to pre-COVID (47.7% and 10.3%, respectively; p: < 0.001). Referral from another hospital was remarkably less during COVID, accounting for 4.5% compared to 27.6% pre-COVID (p < 0.001). Orientation with therapeutic window secondary to previous experience was greater pre-COVID (10.3%) compared with during COVID (4.5%). The commonest cause for early arrival was the patient's previous experience with stroke. This accounted for approximately 25% of cases in both time periods (Table 4).

## Correlation between onset to door time versus method of transportation, education and admission NIHSS

Although the correlation was not significant, transportation by private cars showed a lower onset to door time than that by ambulance (Table 5). The level of education of care giver or patient showed no correlation with onset to door time, although there was a tendency toward being lower if there was postgraduate education. Finally, NIHSS showed a negative yet non-significant correlation with onset to door time (Table 5).

**Table 5. Correlation between onset to door time versus method of transportation, education, and admission NIHSS.**

| Variables | Onset to door time (min) | P value | |
|---|---|---|---|
| Method of transportation | | | Mann–Whitney Test |
| Private car | 480 (180–1440) | 0.6 | |
| Ambulance | 630 (270–2160) | | |
| Education of care giver | | 0.4 | Kruskal–Wallis Test |
| illiterate | 480 (270–1680) | | |
| read and write | 480 (180–1680) | | |
| high school | 600 (150–1642) | | |
| university | 540 (255–1800) | | |
| postgraduate | 420 (120–720) | | |
| Education of patient | | 0.6 | Kruskal–Wallis Test |
| illiterate | 435 (210–1590) | | |
| read and write | 540 (135–2160) | | |
| high school | 480 (180–1590) | | |
| university | 660 (360–1740) | | |
| postgraduate | 420 (150–720) | | |

## Discussion

This study used a questionnaire, completed by patients or their relatives, to derive causes of delayed versus early hospital arrival; thus, reflecting their perspective of onset to door delay.

Among the important causes of delayed presentation to hospital in the pre-COVID-19 period, were unawareness of urgency of the condition and waiting for symptoms to resolve, together accounting for 37.7%. A previous Egyptian study showed that lack of patients' or relatives' knowledge about stroke symptoms, or about the availability of an acute intervention, accounted for 57% of causes of delay [2].

Another Indian study reported that 52% did not perceive the need for medical attention, and 53.5% waited for symptoms to resolve [12]. Two causes can impact awareness of the seriousness of stroke: the level of education and the influence of media. We found that the least time delay was seen in the group of relatives and patients with postgraduate education. This group showed the minimum onset to hospital time, as low as two hours in some cases. Unexpectedly, despite the variability of educational systems and socioeconomic level of different countries, the frequency of lack of awareness and "wait and watch" was similarly reported by other groups in India, Switzerland, China, Egypt, and the United Kingdom (UK) [12–16].

It might seem counter-intuitive that going to another hospital first accounted for both delayed and early arrival. This is explained by a linguistic difference in the questionnaire where "receiving medical help in another hospital" (Table 4) refers to receiving medical service in stroke non-ready hospitals which caused treatment delay, while "referral from another hospital or physician" (Table 5) denotes immediate referral to our stroke center without attempting any intervention thus saving time. This immediate referral from another health facility resulted in early arrival because our university hospital is located within the reach of several governmental and private hospitals where physicians have been oriented with the urgency of thrombolysis for AIS through communication with our stroke team. "Multiple consultations of stroke non-ready hospitals" were reported as delaying hospital arrival in Egypt [15], as well as visiting the family doctor which caused 24% of delays in Switzerland [13], and seeking advice in another hospital in Turkey [11]. Wake-up stroke was also among the significant causes of delay and accounted for 18% of delay. Song et al (2015) similarly found that early arrival was associated with daytime stroke onset [17].

On the other hand, orientation with stroke because of past experience with stroke and awareness of the therapeutic interventions lead to early arrival. This correlates with previous reports that attribute this to a sense of urgency of the condition because of past experience with stroke [6, 7]. Contrary to that, several other studies found that a history of previous stroke did not result in early presentation to hospital [13, 14, 18, 19].

However, orientation with stroke from different forms of media accounted for only 10% of cases of early arrival in pre-COVID-19 time. This raises the demand on media to have a more effective role in stroke awareness as pointed out by Li et al (2019) who emphasized the importance of media in raising public awareness about stroke [14]. Because of the lack of awareness regarding the urgency of the condition, even the severity of stroke did not seem to drive patients or relatives to seek immediate medical help. NIHSS did not show a significant correlation with onset to door. Other studies reported a significant correlation attributable to the alarming nature of severe stroke symptoms [18].

Regarding mode of transportation to hospital, our findings showed that using private cars for transportation to hospital largely outnumbered the use of ambulance service and usually resulted in earlier arrival. This correlates with other Egyptian studies [15, 20]. Although there was a non-significant difference in time delay between both modes of transportation, we observed that the minimum time needed by private car versus ambulance was 3 hours versus

4.5 hours, which infers that using an ambulance can result in arrival to hospital beyond the therapeutic window. Contrary to our findings, several other studies found that transportation by ambulance service was more common and was associated with earlier arrival [13, 17, 21]. This may reflect the perception of the Egyptian public as it relates to the ambulance service being inefficient or unreliable.

When comparing the causes of delayed arrival before and during COVID-19, two new causes emerged: fear of contracting infection and lock down issues. This was reflected as increased stroke severity on presentation during COVID time, since less severe cases might refrain from presenting to hospital as has been previously reported [22]. However, these new causes of delayed presentation did not negatively impact the total number of stroke cases admitted to our center. This differs from other studies from China and New York in which all acute emergencies, including stroke cases, have declined out of fear of contracting infection [23, 24]. In this context, it is worth mentioning that our cases were recruited until May 10. At that time, the incidence curve for COVID-19 cases in Egypt was not at the peak.

On the other hand, lock down had a positive impact on the onset to door time, which was slightly less during COVID-19. This might be because of the availability of relatives and low traffic during lock down. This consequently led to an increased frequency of reperfusion therapy (thrombolysis, thrombolysis with thrombectomy and thrombectomy) by 8.8% during COVID-19 time.

Regarding the various reasons for early hospital arrival, familiarity with stroke symptoms from the media strikingly increased to 47.7% during COVID-19 time, with an increase of 37.4% more than before. Most media highlighted the possible relationship between COVID-19 and stroke, which might have increased public awareness of stroke symptoms. Also, among causes of early arrival was that referrals from other non-stroke ready hospitals significantly decreased as patients came directly to our center instead of seeking help elsewhere, as most governmental hospitals were publicly announced as COVID-19 hospitals. Familiarity with stroke because of previous experience seems to be a constant cause of early presentation, accounting for approximately 25% of causes in both time periods.

## Conclusion

Pre-hospital delay secondary to misperception of the urgency of stroke and management in non-stroke ready hospitals reflected a lack of public and medical staff awareness. This concept is emphasized by findings that previous experience with stroke prompted an early arrival, and by the pronounced effect of media on awareness in the time of COVID-19.

## Recommendations

A media campaign is required to raise public awareness of the urgency of stroke symptoms and of the availability of effective intervention during a specific time window.

In order to respond more promptly and effectively within the treatment window of AIS, stroke centers need to incorporate the Egyptian ambulance services in the management pathway of acute stroke. This can be achieved by educational programs, as well as by implementing a telecommunication system between the ambulance services and hospitals with specialized stroke services.

## Strengths and limitations of the study

Our study has some strengths and limitations. The strengths include the prospective design. This is the first study that spans two different lifetime conditions i.e., pre- and during COVID-

19 pandemic. Also, the study was performed in two accredited stroke centers that serve a large catchment area with all therapeutic facilities.

Among the limitations is that the original study design was changed because of the outbreak of COVID-19; therefore, we had to divide the patients into two groups and study each separately. This resulted in a relatively small number of patients in each group. Also, our findings cannot be generalized because the period of study was short, spanning only five months. However, we found it plausible to study the early impact of COVID-19 on stroke in the first few months of the epidemic.

## Supporting information

**S1 Data. Master sheet (minimal data set).**
(XLSX)

## Author Contributions

**Conceptualization:** Hany M. Aref, Hossam Shokri, Tamer M. Roushdy, Nevine M. El Nahas.

**Data curation:** Hany M. Aref, Hossam Shokri, Nevine M. El Nahas.

**Formal analysis:** Hany M. Aref, Hossam Shokri, Tamer M. Roushdy, Fatma Fathalla, Nevine M. El Nahas.

**Funding acquisition:** Nevine M. El Nahas.

**Investigation:** Hany M. Aref, Hossam Shokri, Tamer M. Roushdy.

**Methodology:** Hany M. Aref, Hossam Shokri, Tamer M. Roushdy, Fatma Fathalla, Nevine M. El Nahas.

**Software:** Hossam Shokri, Nevine M. El Nahas.

**Supervision:** Hossam Shokri.

**Writing – original draft:** Hossam Shokri, Tamer M. Roushdy, Fatma Fathalla, Nevine M. El Nahas.

**Writing – review & editing:** Hossam Shokri, Tamer M. Roushdy, Fatma Fathalla, Nevine M. El Nahas.

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
