## [Decision Letter · Decision Letter 0]

16 Mar 2021

PONE-D-21-03769

Pre-hospital causes for delayed arrival in acute ischemic stroke before and during the COVID-19 pandemic: A study at two stroke centers in Egypt

PLOS ONE

Dear Dr. Shokri,

Thank you for submitting your manuscript to PLOS ONE. After careful consideration, we feel that it has merit but does not fully meet PLOS ONE’s publication criteria as it currently stands. Therefore, we invite you to submit a revised version of the manuscript that addresses the points raised during the review process.

We look forward to receiving your revised manuscript.

Kind regards,

Miguel A. Barboza, MD, MSc

Academic Editor

PLOS ONE

Journal Requirements:

4. Please include your tables as part of your main manuscript and remove the individual files. Please note that supplementary tables should be uploaded. as separate "supporting information" files.

5. Please upload a copy of Supporting Information "supplementary material" which you refer to in your text on page 4.

Reviewers' comments:

Reviewer's Responses to Questions

**Comments to the Author**

1. Is the manuscript technically sound, and do the data support the conclusions?

Reviewer #1: Yes

Reviewer #2: Partly

2. Has the statistical analysis been performed appropriately and rigorously? 

Reviewer #1: No

Reviewer #2: Yes

3. Have the authors made all data underlying the findings in their manuscript fully available?

Reviewer #1: Yes

Reviewer #2: Yes

4. Is the manuscript presented in an intelligible fashion and written in standard English?

Reviewer #1: Yes

Reviewer #2: Yes

5. Review Comments to the Author

Reviewer #1: It was my pleasure to read and review the manuscript: “Pre-hospital causes for delayed arrival in acute ischemic stroke before and during the COVID-19 pandemic: A study at two stroke centers in Egypt” a paper where the different causes of delayed arrival of patients with acute ischemic stroke during a five-month period pre and during de COVID-10 pandemic were analyze and compared.

Before considering this manuscript for publication, I have some comments to add:

Abstract:

Nothing to add

Introduction:

I understand the current situation of pre-hospital care in terms of acute ischemic stroke in Egypt, with the current introduction, but all these paragraphs could be summarized in two, giving special attention to all variables related to patients’ outcome after acute care.

Also, seems relevant to include information regarding the situation of stroke (or medical emergencies) delays after the outbreak of COVID-19 Pandemic in your region or related publications in this topic, as you are comparing both periods, and I cannot see the justification of this behavior in the rationale of your study. This also should be clearly stated in the last paragraph, when adding the main objective of the study.

Methods:

- You have to check the dates in the manuscript because you write that the first COVID’19 case in Egypt was “February 2019”, I assume this was a mistake.

-I suggest to include inclusion and exclusion criteria of your study.

- No information regarding IRB approval protocols are included in your manuscript.

- The survey is a critical part of your study in terms of plausible explanations for delay causes for acute management. I suggest to include the variables and questions for the survey ( a table could be adequate, in this part, instead in the supplementary section).

- Statistical analysis: I suggest to clarify outcome variables for your study, and those included as covariates.

Results:

-I suggest to unify table 1 and 2; both can include the basic variables for the pre-pandemic and trans-pandemic periods

-The section Causes of early hospital arrival within window before and during COVID-19 pandemic is not clear; I can’t find a clear definition for the word “orientation” in terms of the explanation for causes related to early arrival. I suggest to rephrase this section, as I couldn’t understand it.

-I can’t find a clear association among level of education and NIHSS severity…. Is this a relevant finding?? Or seems like a spurious association from the statistical analysis; if you consider this relevant, I suggest to justify a lot in the discussion section, if not, these results seem irrelevant.

Discussion:

-No clear explanation according to previous literature was done in terms of awareness of stroke as an explanatory variable for delay to consultation, at least from what I can read in this section; this should be analyzed since the first approach from your survey, and who is the responsible for this “not adequate awareness” if present.

-Many sections of the discussion are isolated, and no clear association in terms of explanation for the finding on each section were done. I suggest to follow the same order from your paragraphs, when discussing your findings and respective literature support (references).

-You should add a paragraph stating the limitations of your study.

Reviewer #2: Previous question:

The sample size is quite small and have different analyses that could be ineffective for the study objective .

However is a big effort and a good article for describe the barriers, also is a short period to determinate real differences with the pandemic period.

1. Abstract: Methods: it is important to mention wich period of the year, because not all months have the same number of cases of covid-19

Conclusion:

About this subject

Oriented with therapeutic window from media (type: TV. Radio. Social media, newspaper)) pre covid 3.4% during covid 2.3% valor p 0.5

The conclusion said pronounced effect of media in the time of COVID-19. However in the table 4, during COVID was less than the period before -COVID

2.The result should be given in both modalities (Number and Percentages)

3. The analysis about the distance from the hospital should be given in two separates groups, as the methods said with differences between delayed arrival and early arrival and explore statistical significance.

4. In the results:

Comparison of patients’ characteristics before COVID-19 and during COVID-19 time:

- The median time onset door was less, but should said if this time difference was statistical significance

Causes of early hospital arrival within window before and during COVID-19 pandemic:

- Had orientation window therapeutic a statistical significances?

Discussion:

"The type of management showed no significant difference between both time periods. However, it is noticeable that reperfusion therapy (thrombolysis, thrombolysis with thrombectomy and thrombectomy) increased by 8.8% during COVID-19 time, probably due to earlier arrival."

This paragraph is unclear and they don´t mention before any analysis before about treatment. Also the early arrival was 73 patients wich is less than delayed.

References:

References are incomplete. an has different format

Example Reference number 5

6. PLOS authors have the option to publish the peer review history of their article (what does this mean?). If published, this will include your full peer review and any attached files.

Reviewer #1: No

Reviewer #2: **Yes: **Diana Manrique-Otero

---

## [Author Response · Author response to Decision Letter 0]

8 Apr 2021

Review by Plos 1

Pre-hospital causes for delayed arrival in acute ischemic stroke before and during the COVID-19 pandemic: A study at two stroke centers in Egypt

Authors: we would like to express our gratitude to the reviewers for a meticulous revision and succinate corrections that would make the article much more sound. Here we give answers to all of their valuable comments which were all considered and highlighted yellow in the article.

Reviewer #1: It was my pleasure to read and review the manuscript: “Pre-hospital causes for delayed arrival in acute ischemic stroke before and during the COVID-19 pandemic: A study at two stroke centers in Egypt” a paper where the different causes of delayed arrival of patients with acute ischemic stroke during a five-month period pre and during de COVID-10 pandemic were analyze and compared.

Before considering this manuscript for publication, I have some comments to add:

Abstract:

Nothing to add

Introduction:

I understand the current situation of pre-hospital care in terms of acute ischemic stroke in Egypt, with the current introduction, but all these paragraphs could be summarized in two, giving special attention to all variables related to patients’ outcome after acute care. 

The paragraphs were summarized

Also, seems relevant to include information regarding the situation of stroke (or medical emergencies) delays after the outbreak of COVID-19 Pandemic in your region or related publications in this topic, as you are comparing both periods, and I cannot see the justification of this behavior in the rationale of your study. This also should be clearly stated in the last paragraph, when adding the main objective of the study.

Two references added 

Methods:

- You have to check the dates in the manuscript because you write that the first COVID’19 case in Egypt was “February 2019”, I assume this was a mistake. 

This was corrected. The first case was actually announced 15th February 2020 not 2019. 

-I suggest to include inclusion and exclusion criteria of your study. Done

- No information regarding IRB approval protocols are included in your manuscript. Added

- The survey is a critical part of your study in terms of plausible explanations for delay causes for acute management. I suggest to include the variables and questions for the survey ( a table could be adequate, in this part, instead in the supplementary section). 

Added as a figure.

- Statistical analysis: I suggest to clarify outcome variables for your study, and those included as covariates. 

done

Results:

-I suggest to unify table 1 and 2; both can include the basic variables for the pre-pandemic and trans-pandemic periods. done

-The section Causes of early hospital arrival within window before and during COVID-19 pandemic is not clear; I can’t find a clear definition for the word “orientation” in terms of the explanation for causes related to early arrival. I suggest to rephrase this section, as I couldn’t understand it. Clarified 

-I can’t find a clear association among level of education and NIHSS severity…. Is this a relevant finding?? Or seems like a spurious association from the statistical analysis; if you consider this relevant, I suggest to justify a lot in the discussion section, if not, these results seem irrelevant. 

We agree that NIHSS is actually irrelevant to level of education. The correlation in table 4 is between onset to door time versus method of transportation, education and admission NIHSS. Maybe we need to re-phrase the table title to be ‘Correlation between onset to door time and each of method of transportation, education and admission NIHSS.’ if you find this clearer we can change the table title.

Discussion:

-No clear explanation according to previous literature was done in terms of awareness of stroke as an explanatory variable for delay to consultation, at least from what I can read in this section; this should be analyzed since the first approach from your survey, and who is the responsible for this “not adequate awareness” if present.

 A previous study conducted in Egypt for the causes of delayed acute treatment was added.

-Many sections of the discussion are isolate, and no clear association in terms of explanation for the finding on each section were done. I suggest to follow the same order from your paragraphs, when discussing your findings and respective literature support (references). 

Editing of discussion was revised to be more coherent.

-You should add a paragraph stating the limitations of your study. Limitations added.

Reviewer #2: Previous question:

The sample size is quite small and have different analyses that could be ineffective for the study objective .

However is a big effort and a good article for describe the barriers, also is a short period to determinate real differences with the pandemic period.

1. Abstract: Methods: it is important to mention wich period of the year, because not all months have the same number of cases of covid-19 .

The time was specified in the methods section of the body, we also added it also to the abstract.

Conclusion:

About this subject

Oriented with therapeutic window from media (type: TV. Radio. Social media, newspaper)) pre covid 3.4% during covid 2.3% valor p 0.5

The conclusion said pronounced effect of media in the time of COVID-19. However in the table 4, during COVID was less than the period before -COVID

This conclusion about “the pronounced effect of media” refers to the effect of media on orientation with stroke symptoms (not therapeutic window) that was 10.3% pre and 47.7% during COVID; p=<0.001. But we agree that there was no significant effect of media on awareness about therapeutic window. Actually, during COVID the media started to relate vascular disorders, including stroke, to COVID complications, that’s why the public became aware of symptoms rather than rtPA.

2.The result should be given in both modalities (Number and Percentages) done

3. The analysis about the distance from the hospital should be given in two separates groups, as the methods said with differences between delayed arrival and early arrival and explore statistical significance. Added to results

Early arrival median distance (IQR)= 20 (10-30), late arrival median distance (IQR)= 20 (10-35) (p value= 0.12) 

4. In the results:

Comparison of patients’ characteristics before COVID-19 and during COVID-19 time:

- The median time onset door was less, but should said if this time difference was statistical significance

 the p value is O.29 and highlighted yellow in table 1

Causes of early hospital arrival within window before and during COVID-19 pandemic:

- Had orientation window therapeutic a statistical significances? It was not statistically significant p=0.5. highlighted yellow in table 3

Discussion:

"The type of management showed no significant difference between both time periods. However, it is noticeable that reperfusion therapy (thrombolysis, thrombolysis with thrombectomy and thrombectomy) increased by 8.8% during COVID-19 time, probably due to earlier arrival." 

This paragraph is unclear and they don´t mention before any analysis before about treatment. 

In the results section we mentioned “Total cases receiving reperfusion therapy were more during COVID 25.7% versus 17.1% in pre-COVID time” (highlighted yellow). the difference between them is 8.8% as stated in discussion.

Also the early arrival was 73 patients wich is less than delayed. 

Here we do not refer to the total number of patients who arrived early, but we mean that arrival was earlier due to shorter onset to door during COVID (median: 480 min ) than before COVID (median: 540 min) which might explain the higher frequency of reperfusion therapy. 

References:

References are incomplete. an has different format

Example Reference number 5 corrected

---

## [Editor Report · Decision Letter 1]

17 May 2021

PONE-D-21-03769R1

Pre-hospital causes for delayed arrival in acute ischemic stroke before and during the COVID-19 pandemic: A study at two stroke centers in Egypt

PLOS ONE

Dear Dr. Shokri,

Thank you for submitting your manuscript to PLOS ONE. After careful consideration, we feel that it has merit but does not fully meet PLOS ONE’s publication criteria as it currently stands. Therefore, we invite you to submit a revised version of the manuscript that addresses the points raised during the review process.

The authors addressed all the requested comments and suggestions from reviewers, but there are some minor issues that should be evaluated by the authors before considering this manuscript suitable for publication

***1. There are several grammatical mistakes throughout the manuscript, therefore English editing services are highly required.***

***2. When reading the abstract, Background doesn't reflect the purpose of the study; if you are stating that the rationale of the study is to evaluate causes for pre-hospital delay in acute stroke cases seeking for acute revascularization, knowing that your group previously studied this condition seem interesting for the introduction section, but does not explain the reason why you are performing the study. I suggest to re-write te background, and also CLEARLY state that the objective of the present study is to investigate the above-mentioned delay BEFORE and AFTER the COVID-19 outbreak.***

***3. Please adhere to Plos One guidelines in terms of tables and figures. Delete colors from columns and rows, and explain abbreviations used. Please erase final markup inside table's text (see table 2)***

***4. P value in table 2 refers to...?? There several variables compared among themselves, and only a P value, and I cannot see the correspondance of it; also, if you are intending to show association among both periods, I think that you should add p values comparing each one of the variables***

***5. Table 4: I feel that the last column is irrelevant; you could add a superscript explaining the statistical analysis you used. Admission NIHSS result in this table seems odd; I can't understand what you are intending to prove if the comparison is Onset-to-door time and NIHSS... If you are proving that higher of lower NIHSS could increase the delay in terms of consultation, this analysis should be better in a scatter plot, outside this table; if you intend to analyze NIHSS as categories, you should explain since the Methods section, how you manage this variable; as the correlation was negative, I see irrelevant to add this information in a table or a figure.***

***6. I don't understand the reason why the questionnaire was generated as a figure; It's a table, and this could be sent as a normal Word format document. ***

We look forward to receiving your revised manuscript.

Kind regards,

Miguel A. Barboza, MD, MSc

Academic Editor

PLOS ONE
---

## [Author Response · Author response to Decision Letter 1]

28 May 2021

1. There are several grammatical mistakes throughout the manuscript, therefore English editing services are highly required.

Done 

2. When reading the abstract, Background doesn't reflect the purpose of the study; if you are stating that the rationale of the study is to evaluate causes for pre-hospital delay in acute stroke cases seeking for acute revascularization, knowing that your group previously studied this condition seem interesting for the introduction section, but does not explain the reason why you are performing the study. I suggest to re-write te background, and also CLEARLY state that the objective of the present study is to investigate the above-mentioned delay BEFORE and AFTER the COVID-19 outbreak.

Done 

3. Please adhere to Plos One guidelines in terms of tables and figures. Delete colors from columns and rows, and explain abbreviations used. Please erase final markup inside table's text (see table 2)

Done 

4. P value in table 2 refers to...?? There several variables compared among themselves, and only a P value, and I cannot see the correspondance of it; also, if you are intending to show association among both periods, I think that you should add p values comparing each one of the variables

Done 

5. Table 4: I feel that the last column is irrelevant; you could add a superscript explaining the statistical analysis you used. Admission NIHSS result in this table seems odd; I can't understand what you are intending to prove if the comparison is Onset-to-door time and NIHSS... If you are proving that higher of lower NIHSS could increase the delay in terms of consultation, this analysis should be better in a scatter plot, outside this table; if you intend to analyze NIHSS as categories, you should explain since the Methods section, how you manage this variable; as the correlation was negative, I see irrelevant to add this information in a table or a figure.

Done 

6. I don't understand the reason why the questionnaire was generated as a figure; It's a table, and this could be sent as a normal Word format document. 

Done

---

## [Editor Report · Decision Letter 2]

23 Jun 2021

Pre-hospital causes for delayed arrival in acute ischemic stroke before and during the COVID-19 pandemic: A study at two stroke centers in Egypt

PONE-D-21-03769R2

Dear Dr. Shokri,

We’re pleased to inform you that your manuscript has been judged scientifically suitable for publication and will be formally accepted for publication once it meets all outstanding technical requirements.

Kind regards,

Miguel A. Barboza, MD, MSc

Academic Editor

PLOS ONE
---

## [Editor Report · Acceptance letter]

1 Jul 2021

PONE-D-21-03769R2 

Pre-hospital causes for delayed arrival in acute ischemic stroke before and during the COVID-19 pandemic: A study at two stroke centers in Egypt 

Dear Dr. Shokri:

I'm pleased to inform you that your manuscript has been deemed suitable for publication in PLOS ONE. Congratulations! Your manuscript is now with our production department. 

Kind regards, 

on behalf of

Dr. Miguel A. Barboza 

Academic Editor

PLOS ONE